# Assessment of a decentralization model in improving treatment and care of visceral leishmaniasis in Turkana County, Kenya: A mixed method study

Jane Mbui[1], Mariam Macharia[2], Dawn Maranga[3], Collins Okoyo [2,4]*

1 Centre for Clinical Research (CCR), Kenya Medical Research Institute (KEMRI), Nairobi, Kenya, 2 Eastern and Southern Africa Centre of International Parasite Control (ESACIPAC), Kenya Medical Research Institute (KEMRI), Nairobi, Kenya, 3 Neglected Tropical Diseases Programme (NTDP), Foundation for Innovative New Diagnostics (FIND), Nairobi, Kenya, 4 Department of Epidemiology, Statistics and Informatics (DESI), Kenya Medical Research Institute (KEMRI), Nairobi, Kenya

* collinsomondiokoyo@gmail.com

## Abstract

### Background

Visceral Leishmaniasis (VL) is a vector-borne disease caused by the protozoa *Leishmania* and transmitted by sandflies. Up to 60% of all VL cases worldwide occur in East Africa. Given its ranking as a main cause of death among the parasitic infections worldwide, VL constitutes a serious global health concern. In Turkana County, the Foundation for Innovative New Diagnostics has undertaken significant work in supporting the decentralization of access to diagnosis and treatment, expanding from six health facilities giving VL care in 2018 to twenty-two in 2022. This study sought to assess the decentralization of VL services in Turkana to inform policy at the county and national levels.

### Methods

This was a mixed methods cross-sectional survey conducted in four selected health facilities within Turkana County, between November 2023 and February 2024. Quantitative data involved data abstraction from records of VL patients between January 2018 to December 2022. For the qualitative data, 13 in-depth interviews were conducted with VL patients, 16 key informant interviews (KIIs) with healthcare workers, and seven KIIs with the county health management team members. Descriptive analysis of the quantitative data and thematic analysis of qualitative data were performed to assess the decentralization model in improving VL treatment and care in Turkana County.

**Data availability statement:** All relevant data supporting the conclusions of this paper are provided within the article. The raw datasets used in the analyses can be accessed in the GitHub public repository via the link: https://github.com/mancollo/VL-Decentralisation-Paper.

**Funding:** This work was funded by the European and Developing Countries Clinical Trials Partnership (EDCTP2) programme, supported by the European Union (Grant No. RIA2020S-3301), through the Drugs for Neglected Diseases initiative (DNDi) and the Foundations for Innovative New Diagnostics (FIND). The funders had no role in study design, data collection and analysis, decision to publish, or preparation of the manuscript.

**Competing interests:** The authors have declared that no competing interests exist.

## Results

The community had low knowledge of VL signs and symptoms. The mean delay time since the onset of symptoms before seeking medical care was 46.9 days. This long delay was mainly attributed to the long distance to the health facilities and the high costs of accessing the treatment facilities. Majority of the patients sought traditional treatments first before visiting the health facilities. Further, health workers indicated that the decentralization model has led to accurate diagnosis of VL and improvement of the infrastructure within the health facilities.

## Conclusion

The study observed low awareness of VL disease among patients that contributed heavily to delayed time to diagnosis. This calls for revamped health education and awareness campaigns among the communities living in VL endemic areas to promote positive behaviour change for effective control and elimination of the disease.

## 1. Introduction

Leishmaniasis is a vector-borne disease caused by the protozoa *Leishmania* and transmitted by female phlebotomine sandflies. It manifests in three main forms: Visceral leishmaniasis (kala-azar), Cutaneous leishmaniasis, and Mucocutaneous leishmaniasis [1]. The disease spectrum ranges from a localized skin infection (cutaneous) to a potentially fatal visceral form. The number of new cases of visceral leishmaniasis (VL) per year is estimated to lie between 50,000–90,000 cases globally, with a case fatality ratio exceeding 95% when left untreated [2]. Given its ranking as a major cause of death, among parasitic diseases, worldwide with epidemic potential, VL constitutes a severe global health concern [3]. Despite increased public interest in VL control, with the World Health Organization (WHO) aiming to eliminate VL by the year 2030, the number of VL patients remains largely stable, with only a slight decline in recent years [4,5].

East Africa is currently the most affected region globally, accounting for 60% of VL cases [6]. Endemic foci are found in remote areas, in Kenya, Somalia, South Sudan, Sudan, Ethiopia, and Eritrea. Due to population displacements during conflicts, endemic areas are continually expanding, which results in naive populations entering endemic areas where they are at high risk of VL infection. This population movement of people from non-endemic areas may result in large-scale outbreaks, as has been observed in Ethiopia, Kenya, and South Sudan [7]. In Kenya, VL is endemic in 11 counties namely, Turkana, Baringo, West Pokot, Isiolo, Marsabit, Kajiado, Kitui, Tharaka Nithi, Garissa, Wajir, and Mandera. The estimated case burden is around 2,000 cases annually [8].

VL has a broad range of symptoms, with disease manifestations ranging from asymptomatic to life-threatening cases. The most common symptoms include fever, weight loss, wasting, and hepatosplenomegaly. Risk factors for the disease include immunosuppressive illnesses such as human immunodeficiency virus (HIV) infection

and organ transplantation [9,10]. In Kenya, VL is a paediatric disease with children below 14 years accounting for 60% of the total patients, males are disproportionately affected compared to females [9]. VL is diagnosed using a standard diagnosis algorithm as contained in the Ministry of Health VL diagnosis and case management guidelines of 2017 [12]. The algorithm includes a clinical case definition in combination with the laboratory-based tests, such as rK39 rapid diagnostic test (RDT), direct agglutination test (DAT), microscopy, culture, and molecular tests [12]. The standard first-line treatment of VL in East Africa is a 17-day combination of sodium stibogluconate (SSG) and paromomycin (PM) therapy, an improvement over the previous 30-day SSG monotherapy. The follow-up of treated VL cases is conducted six months after treatment completion [11]. The VL control is challenging in East Africa due to the absence of a highly sensitive rapid test for diagnosis, a complex treatment regimen that can only be implemented at a well-equipped health facility, and the absence of measures to reduce the sand fly vector population. The control is further compounded by populations at risk living in poverty, residing in remote areas that are far from hospitals, and insecurity-prone due to recurrent internal conflicts [12].

Although diagnostic tests and treatment for leishmaniasis are free in Kenya, barriers to access to care continue to constitute a major threat to disease control and elimination. These barriers lead to delays in diagnosis and treatment, resulting in higher mortality rates. The barriers hinder the achievement of the elimination threshold target set by WHO and contribute to continued persistent disease transmission in the community. Decentralization of diagnosis and treatment services has shown potential health benefits in addressing the leishmaniasis disease burden [13].

In Turkana County, the Foundation for Innovative New Diagnostics (FIND) has undertaken significant work in supporting the decentralization of access to diagnosis and treatment, expanding from six health facilities giving VL care in 2018 to a total of twenty-two facilities in 2022. In this context, decentralization of access to care means the expansion of health care facilities that can provide VL diagnosis and/or treatment. Turkana is among the first counties endemic for VL in Kenya to implement this decentralization model.

This study aimed to assess the effects of an expanded access model (decentralization) of VL care, diagnosis and treatment, and to understand the barriers and facilitators of the implementation of this model in Turkana County, Kenya. The study sought to answer the following research questions; i) what is the total number of people with VL symptoms tested and confirmed to have VL, ii) what is the time to diagnosis and initiation of treatment?, iii) what is the proportion of VL patients completing treatment and achieving initial cure?, iv) what is the knowledge and practices of patients about VL disease?, v) what is the knowledge of healthcare providers in managing VL disease?, and, vi) what are the barriers and facilitators to VL treatment and care?

## 2. Materials and methods

### 2.1 Study design

This was a cross-sectional descriptive mixed methods study. Quantitative data were abstracted from health records of VL patients from 2018 to 2022 in four selected health facilities. This was to assess whether there were trends in patient turnaround time since the decentralization process began. All health records of VL patients were reviewed, data abstracted and entered into the Research Electronic Data Capture (REDCap) system, an electronic mobile-based data collection platform. For the qualitative data, in-depth interviews were held with patients or their primary caregivers for patients below 18 years old who were receiving treatment for VL in these four facilities during the data collection phase of the study. Additionally, key informants' interviews were conducted with selected healthcare workers within the same facilities and select members of the County Health Management Team (CHMT).

### 2.2 Study site

The study site was Turkana County, which was purposively selected because it is implementing the VL decentralization model. Within the county, four health facilities were selected; Lodwar County Referral Hospital (LCRH), Namoruputh PAG Health Center (NPHC), Lopiding Sub-County Hospital (LSCH), and Kakuma Sub-County Hospital (KSCH) (Fig 1).

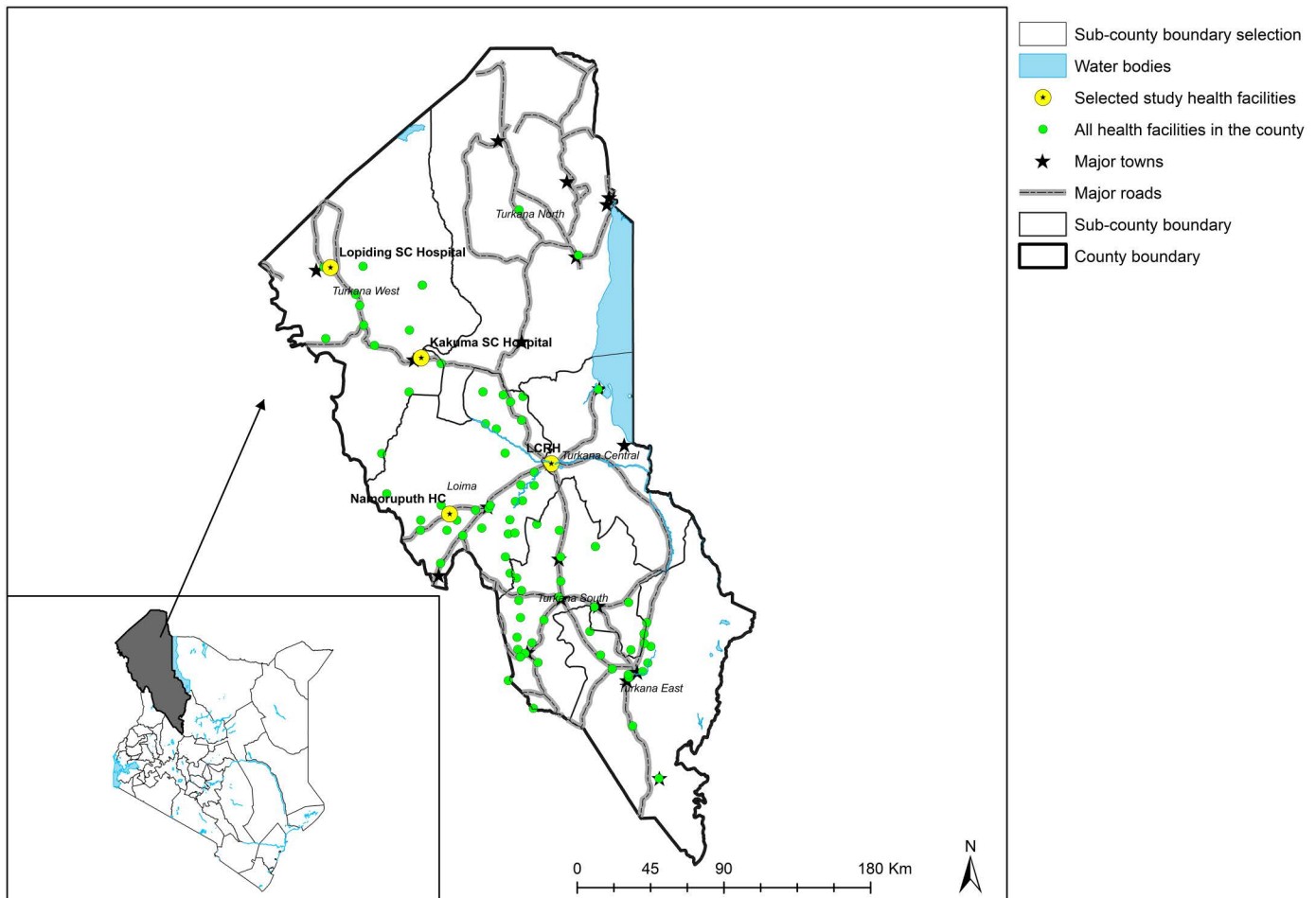

**Fig 1. A map of Turkana County showing all the health facilities in the county and the selected health facilities where the study was conducted.** The selected facilities are part of the facilities that have been participating in the decentralization model. The map was created using ArcGIS Desktop version 10.2.2 software (Environmental Systems Research Institute Inc., Redlands, CA, USA). The shapefile for the county was obtained from the Regional Centre for Mapping of Resources for Development (RCMRD) (https://opendata.rcmrd.org), and the base layer of the map was obtained from Environmental Systems Research Inc. (ESRI) (https://www.esri.com/en-us/arcgis/products/arcgis-online/features/make-maps).

These facilities were purposefully selected for having VL patients under treatment during the data collection period. Notably, three of these hospitals (LCRH, LSCH, and KSCH) are public health facilities under the county government of Turkana, while NPHC is a faith-based private health facility supported by the Pentecostal Assemblies of God (PAG) church.

## 2.3 Study population

For the quantitative data, the population included records of patients of all ages who were diagnosed with VL from the four health facilities between 2018 and 2022. It also included the monthly summaries of VL cases and stock management forms. For the qualitative data, the population included VL patients of all ages who were receiving treatment from the four health facilities during the data collection period, healthcare providers working in these facilities (clinicians, nurses, laboratory technologists, and pharmacists), and members of the CHMT.

## 2.4 Sample size

The study abstracted all VL-related data from four health records for the quantitative data namely, (i) case management form, (ii) laboratory log, (iii) monthly summary of VL cases, and (iv) monthly summary of VL commodity stock management. These records were accessed from patients diagnosed with VL on diverse dates between 4th January 2018 and 29th December 2022. For the qualitative data, a total of 13 in-depth interviews (IDIs) were conducted with purposively selected patients and caregivers of children with VL within the four facilities. Further, 16 key informant interviews (KIIs) were conducted with purposively selected healthcare workers within the facilities, and another seven KIIs were held with members of the CHMT.

## 2.5 Data collection

Both quantitative and qualitative data were collected between November 2023 and February 2024. The quantitative data abstraction was done between 9th and 27th February 2024. The team engaged the facility health records officers who abstracted data from VL patients' records, and captured the data electronically using the REDCap, a data collection and management platform. The qualitative data was collected between 27th November and 15th December 2023. The team trained four research assistants who were conversant with the local language. Structured IDI and KII guides were used to moderate the interviews. The guides contained questions on participants' knowledge, health-seeking behaviors, barriers and facilitators to VL care and perceptions. Interviews with the patients and healthcare workers were conducted within the health facility, while interviews with the members of the CHMT were conducted in their respective offices, with a few who were not in the office conducted via telephone interviews. All interviews were audio recorded using digital voice recorders.

## 2.6 Data management and analysis

For the quantitative data, data was abstracted from participants' health records and entered into REDCap, which incorporated in-built data quality checks to prevent errors. The entered data was transmitted to a centralized secure server based in Kenya Medical Research Institute, using android-based mobile devices. The data was then downloaded into Microsoft Excel files and imported into STATA version 18.0 (STATA Corporation, College Station, TX, USA) for statistical analysis. Descriptive analyses were performed for various indicators that included; time to diagnosis, time to treatment after diagnosis, treatment completion rates, and treatment outcomes. For the qualitative data, data was transcribed verbatim, and the scripts translated to English. A second set of two research assistants listened through the interviews to ensure that the English transcript reflected the actual interviews that were conducted in Turkana language (back translation). Once the back translation was completed, the scripts were read by the study investigators, who comprised a social scientist. The final transcripts were organized into Microsoft Word files for manual analysis. Thematic areas were identified, and a coding framework developed (appendix 1). The data was then entered into NVivo version 12 for further thematic analysis. We compared the codes within and across transcripts and summarized coded extracts grouped into themes and sub-themes as appropriate. The analyzed data was then presented in text format [14].

## 2.7 Ethics statement

The ethical approval for this study was obtained from the Amref Health Africa Ethics and Scientific Review Committee (ESRC) under reference number P1476-2023. A research permit No. NACOSTI/P/23/25936 was obtained from the Kenya National Commission for Science, Technology, and Innovation (NACOSTI). Written informed consent was sought from all study participants before the start of the interviews. Written permission to access patients' archived medical records was sought from the heads of the respective health facilities and the County Department of Health only during the period of the study.

## 3. Results

Overall, for the quantitative data, data was abstracted from four types of patient health records, namely, case management form (725 records abstracted), laboratory log (722 records), monthly case summary (150 records), and monthly commodity stock management summary (103 records). The records were abstracted from the four health facilities in Turkana County. The records reviewed were for patients seeking care and treatment for VL on diverse dates between 4th January 2018 and 29th December 2022. Table 1 shows the number of case management records abstracted per facility between 2018 and 2022.

### 3.1 Socio-demographic characteristics

For the quantitative data, the mean age of the patients as documented in the case management form was 14.3 years (SD: 12.0 years, range: 1–72 years). Children aged 14 years and below were 437 (67.9%). A majority, 506 (70.2%), of the patients were males, and 215 (29.8%) were females. From the female patients, pregnancy status was obtained from only 68 patients, where 1 (1.5%) was pregnant at the time of diagnosis, 26 (38.2%) were not pregnant, and 41 (60.3%) had unknown pregnancy status (or pregnancy test not done). Information on patients' occupation was obtained from 706 records, where 458 (64.9%) had no occupation, 180 (25.5%) were animal herders, 46 (6.5%) were students, 12 (1.7%) were homemakers, 9 (1.3%) were self-employed, and 1 (0.1%) had other occupation. Information on the level of education was obtained from 704 records, where the majority 622 (88.4%) had no formal education, 69 (9.8%) had a primary level of education, 8 (1.1%) had secondary, and 5 (0.7%) had post-secondary education.

For qualitative data, a total of thirteen (n = 13) IDIs were conducted with VL patients and the caregivers of minors. Overall, majority (n = 11) of the patients were children aged between 1–16 years, the rest were, one emancipated minor (17 years) and one adult (30 years). Of the eleven caregivers interviewed, over half were female (n = 6), with all the participants having no formal education and unemployed (Table 2).

A total of sixteen (n = 16) KIIs were conducted with the health workers sampled from the four health facilities. There was an equal number of male (n = 8) and female (n = 8) participants. The age of the participants ranged between 27–46 years. All the participants had a tertiary level of education (n = 16). The participants consisted of five nurses, four clinical officers, four pharmacists, and three laboratory technologists.

Additionally, seven members of CHMT participated in the KIIs. All participants were male, aged between 30–40 years with an average of 36.7 years, and the majority had attained a Master's level of education. The majority of these participants had served in their current role for over four years.

### 3.2. Knowledge, signs, and symptoms of visceral leishmaniasis

From the quantitative data, the following signs were evaluated when patients presented to the health facility for diagnosis, weight, height, body temperature, abdominal distension, and liver and spleen enlargement. The patients were further tested

**Table 1. Number of case management records extracted per health facility surveyed in Turkana County between the year 2018 to 2022.**

| Health facility | 2018 | 2019 | 2020 | 2021 | 2022 | Total |
|---|---|---|---|---|---|---|
| Case management record | | | | | | |
| Kakuma SCH | 3 | 5 | 2 | 17 | 9 | 36 |
| Lodwar CRH | MR | MR | MR | 91 | 81 | 172 |
| Lopiding SCH | 12 | 25 | 17 | 43 | 74 | 171 |
| Namoruputh PAG HC | 102 | 48 | 58 | 50 | 88 | 346 |
| **Total** | **117** | **78** | **77** | **201** | **252** | **725** |

MR: Indicates that there were missing records for the specific years in the particular facility

Overall, for qualitative data, data were collected using IDIs conducted with the VL patients and the caregivers of minors. KIIs were conducted with the healthcare workers attending to the VL patients and with the CHMT members responsible for policy decisions at the county level.

**Table 2. Socio-demographic characteristics of participants for the qualitative data collected from participants surveyed in Turkana County between the year 2018 to 2022.**

| Characteristic | n = 13 | Percent (%) |
|---|---|---|
| **Gender of caregiver** | | |
| Male | 5 | 45.5 |
| Female | 6 | 54.5 |
| **Gender of patients** | | |
| Male | 11 | 84.6 |
| Female | 2 | 15.4 |
| **Age of patients in years** | | |
| < 5 | 2 | 15.4 |
| 5-10 | 4 | 30.8 |
| 10-15 | 4 | 30.8 |
| 15-20 | 2 | 15.4 |
| > 20 | 1 | 7.8 |
| **Age of caregivers in years** | | |
| < 25 | 2 | 18.2 |
| 25-35 | 4 | 36.4 |
| 36-45 | 4 | 36.4 |
| > 45 | 1 | 9 |
| **Education status of caregivers** | | |
| No education | 9 | 81.8 |
| Primary incomplete | 2 | 18.2 |
| Primary complete | 0 | 0 |
| **Occupation** | | |
| Unemployed | 6 | 54.5 |
| Housewife | 1 | 9.1 |
| Farmer/pastoralist | 4 | 36.4 |
| Employed | 0 | 0 |

in the laboratory using rapid diagnostic test kits (RDT) for VL and for malaria, and hemoglobin levels were also assessed. Weight information was obtained from 697 records, with a mean weight of 31.9 kg (SD: 16.6 kg, range: 2–120 kg). Information on height was obtained from 368 records, with a mean height of 137.4 cm (SD: 33.4 cm, range: 7–194 cm). Body temperature measurement was recorded in 642 patient files, with a mean temperature of 37.4°C (SD: 1.4°C, range: 23.9-40.0°C). A total of 703 (96.8%) patients reportedly had fever 2 weeks before the diagnosis date, 621 (88.1%) had abdominal distension, 178 (30.2%) had their liver enlarged, and 647 (91.1%) had their spleen enlarged. Measurements for the organ enlargements indicated that the mean liver enlargement was 5.5 cm (SD: 3.4 cm, range: 2–16 cm) and the mean spleen enlargement was 6.5 cm (SD: 3.3 cm, range: 2–22 cm). Information on the presence of concomitant infections and conditions was obtained from 595 records, out of which 124 (20.8%) had other concomitant illnesses like malaria, gastrointestinal infections, malnutrition, pneumonia, respiratory infections, and skin infections, among others. Information on the signs of post-kala-azar dermal leishmaniasis (PKDL) was obtained from 124 records, out of which 1 (0.8%) patient had signs of PKDL.

From the qualitative data, knowledge and awareness about VL were assessed using the validated qualitative guides. The participants were asked about the causes, symptoms, people at risk, severity, and prevention measures of VL. Participants indicated that VL is commonly referred to as "kala-azar" or "*Etid*" in their community. The participants had little awareness about VL, but they had heard of a community or family member with an enlarged spleen. However, they did not associate the swollen abdomen and other VL symptoms directly with the disease.

*"I can't know because there are some people in the community who have the symptoms of enlargement of the stomach but we didn't know there was such condition, because nobody gives awareness about the disease or the condition that there is kala-azar…."* (IDI_CareGiver_R002_LCRH).

The health workers indicated that most patients come to the facility with high fever, hepatosplenomegaly, abdominal pain, bleeding tendencies, and headache among other symptoms.

*"They come with complaints of fever. They come with complaints of headache, they also have enlarged spleen, some have enlarged "eeh" organs like liver, also some have complaints of nose bleeding and also "aah" some are pale, showing the indication of low. blood levels in their body"* (KII_HCW_R003_NPHC).

### 3.3 Diagnosis of visceral leishmaniasis

From the quantitative data, the mean delay time since the onset of symptoms before seeking diagnosis was reported as 46.9 days (SD: 43.6 days, range: 1–365 days). Patients seeking diagnosis at Kakuma SCH took slightly longer than patients attending other facilities (Table 3). Overall, 718 (99.0%) of the patients were diagnosed with primary VL, 5 (0.7%) were relapses, and 2 (0.3%) were diagnosed with other conditions. The main method of diagnosis was by RDT (78.4%), followed by direct agglutination test (DAT) (21.4%).

From the qualitative data, the patients and caregivers indicated that they did not seek medical care immediately after the onset of symptoms. The waiting period varied between two weeks up to about one year with majority taking over one month before visiting the hospital (Table 4). The mentioned reasons for the delay included the long distance to the nearest health facility that can diagnose and treat VL, high cost of transport to the facility, and the initial use of traditional forms of treatment.

*"Kala-azar is a swollen abdomen and when it's cut and blood pours, it subsides that's why I waited "greets a friend". When the cutting failed is when I brought the child to the hospital. The facility is far from home, and there is lack of transport to reach the facility. Livestock is being sold to get money for transport using motorbikes, that's the only source of transport here to Namoruputh."* (IDI_CareGiver_R006_NPHC).

According to the health workers, most of the patients took a long time on average of two months to seek medical attention. The patients only came when their health had deteriorated and even developed complications. They cited several reasons that included long distances to the facility, lack of transport, using traditional treatment methods before seeking

**Table 3. Number of days, mean (min-max), taken to seek diagnosis since the onset of the symptoms from the health facilities surveyed in Turkana County between the year 2018 to 2022.**

| Health facility | Overall | Mean number of days since onset of symptoms per year | | | | |
|---|---|---|---|---|---|---|
| | | 2018 | 2019 | 2020 | 2021 | 2022 |
| Overall | 46.9 (1-365) | 33.3 (14-150) | 45.2 (3-240) | 43.8 (10-180) | 55.0 (1-365) | 47.7 (1-365) |
| Facility | | | | | | |
| Kakuma SCH | 53.6 (21-90) | 45.7 (21-60) | 67.8 (21-90) | 30 (30-30) | – | – |
| Lodwar CRH | 49.1 (1-365) | MR | MR | MR | 46.1 (1-365) | 48.9 (2-150) |
| Lopiding SCH | 45.9 (1-365) | 28.7 (14-60) | 36.8 (10-240) | 41.4 (10-180) | 58.9 (1-365) | 45.1 (1-365) |
| Namoruputh PAG HC | 46.1 (3-210) | 33.5 (14-150) | 47.1 (3-180) | 45.0 (14-150) | 67.4 (14-168) | 48.7 (9-210) |

- Indicates that the indicator of interest was not available at the specific facility for that year

MR: Indicates that there were missing records for the specific years in the particular facility

**Table 4. Time duration spent before seeking medical attention, reasons for the delay, and symptoms that made participants seek care in selected health facilities surveyed in Turkana County between the year 2018 to 2022.**

| Participant ID | Time spent before seeking care | Reason for delay | Symptom that made them to seek medical care |
|---|---|---|---|
| 001 | 4 months | Lack of money | Severe body weakness |
| 002 | 1 month | Treated for malaria | Pale eyes |
| 003 | 5 months | Long distance to the facility; lack of money for transport | Fever and body weakness |
| 004 | 3 months | Initial use of traditional treatments | Nose bleeding |
| 005 | 2 weeks | Long distance to the facility; lack of money for transport | Gross weight loss |
| 006 | 2 months | Initial use of traditional treatments | Nose bleeding |
| 007 | 1 year | Initial use of traditional treatments | Nose bleeding |
| 008 | 1 month | Initial use of traditional treatments | General body weakness |
| 009 | 1 month | Treated for malaria | Protruding abdomen |
| 010 | 1 month | Lack of money for transport | High fever |
| 011 | 5 months | Initial use of traditional treatments | Gross weight loss |
| 012 | 3 months | Treated for malaria | Protruding abdomen |
| 013 | 1 month | Lack of awareness of what the child was suffering from | Too much sleep |

conventional treatment, negative attitude towards the hospital due to fear of dying in the hospital, lack of knowledge and awareness of the disease, financial constraints where some have to wait to sell their livestock, and insecurity in the region.

*"I will say "aah" they take maybe like two months "yeah" because you know for...the places they come from, they may try out some traditional interventions like "aah"... bloodletting, where they cut the spleen area. They cut then some blood gets spilled over...so some even go to the extent of butchering a goat, you know, those are traditional interventions. When those interventions don't work, that's when they come to the hospital. They come when the patients are usually weak...so it's an average of two months… "yeah"* (KII_HCW_R007_KSCH).

### 3.4 Treatment and follow-up of visceral leishmaniasis

Of the 725 patients diagnosed with VL, 648 (89.4%) were started on treatment. On average, the patients were initiated on treatment 11.0 days (SD: 63.4 days, range: 0–1,092 days) since diagnosis. On average, patients seeking treatment at Kakuma were initiated on treatment the same day upon diagnosis, patients attending Namoruputh and Lopiding were initiated after 9 days, while those attending Lodwar were initiated after about 19 days (Table 5). Among the patients initiated on treatment, 493 (76.1%) were put on the standard combination therapy of SSG and PM. Of all the patients initiated on treatment, treatment completion data was obtained from 427 (65.9%), out of which 424 (99.3%) reportedly completed treatment. Treatment outcome was recorded among 440 (98.2%) patients, out of which, 434 (98.6%) were initially cured, 2 (0.5%) were confirmed non-response, 1 (0.2%) was a probable non-response, 1 (0.2%) defaulted, and 2 (0.5%) reported deaths resulting from VL. There were no reported severe adverse events from the administered medications. In addition, 240 (57.6%) of the patients were reportedly treated for other concomitant conditions. Of all patients initiated on treatment, only 16 (2.5%) returned for follow-up visits, with all indicating a final definitive cure.

Results from the qualitative data indicated that the drugs available for VL treatment at the health facilities were SSG, PM and ambisome injection. Patients are required to visit the nearest health facilities for follow-up visit. However, due to the long distances, patients rarely turn up. In addition, the health facilities do not have a follow-up mechanism to reach the patients.

**Table 5. Number of days, mean (min-max), taken to initiate patients to treatment since the diagnosis in selected health facilities surveyed in Turkana County between the year 2018 to 2022.**

| Health facility | Overall | Mean number of days since initiation of treatment per year | | | | |
| --- | --- | --- | --- | --- | --- | --- |
| | | 2018 | 2019 | 2020 | 2021 | 2022 |
| Overall | 11.0 (0-1092) | 0.9 (0-13) | 0.7 (0-7) | 11.2 (0-176) | 23.9 (0-1092) | 8.1 (0-375) |
| Facility | | | | | | |
| Kakuma SCH | 0.3 (0-5) | 0 | 2.5 (0-5) | 0 | 0 | 0 |
| Lodwar CRH | 19.7 (0-375) | MR | MR | MR | 17.4 (0-366) | 23.3 (0-375) |
| Lopiding SCH | 9.6 (0-366) | 0.7 (0-2) | 1.0 (0-7) | 1.1 (0-14) | 31.1 (0-366) | 3.7 (0-234) |
| Namoruputh PAG HC | 9.1 (0-1092) | 0.9 (0-13) | 0.4 (0-1) | 14.8 (0-176) | 38.0 (0-1092) | 2.8 (0-33) |

- Indicates that the indicator of interest was not available at the specific facility for that year

MR: Indicates that there were missing records for the specific years in the particular facility

> "*We follow up rarely; rarely do we do the follow-up. When the patient gets healed, we are done. We rarely do the follow-up, but if we get any relapse, maybe someone has been treated for kala-azar and comes back with the same symptoms then we advise for parasitological techniques to rule out if it is VL or other infections that the patient has*" (KII_HCW_R009_LSCH).

### 3.5 Monthly summary of visceral leishmaniasis cases and commodity stock management at the facility level

From the quantitative data, a total of 150 monthly summary records for leishmaniasis cases were abstracted from the four health facilities. According to the facilities, 61 (40.7%) records were abstracted from Namoruputh, 58 (38.7%) abstracted from Lopiding, and 31 (20.7%) abstracted from Kakuma. However, no monthly summary record was obtained from Lodwar. Overall, 1,640 leishmaniasis suspect cases were reported from the four health facilities between 2018 and 2022. Out of these cases, 632 (38.5%) were positive for VL, 7 were VL relapses, and 18 were deaths. According to the years, more positive cases were reported in 2022 (202 cases), followed by 2021 (117 cases). The cases reported according to the health facilities and record abstraction year are shown in Table 6.

Further, a total of 103 records for monthly summaries of leishmaniasis commodity stock management were abstracted from the four facilities. The number of days of stockouts varied per commodity as shown in Table 7. All the facilities did not stock the following commodities, glucantime, miltefosine, and DAT freeze-dried, with longer stockout period reported for PM and SSG.

From the qualitative data, stock management is mainly divided into two, diagnosis and treatment, and is done as per department using stock cards. The county and sub-county laboratory coordinators are in charge of supplying the testing kits to the health facility. The drugs are coordinated by the county and sub-county pharmacists, through the neglected tropical diseases (NTD) coordinator, who ensures that drugs are supplied to the pharmacist at the health facilities. In instances where shortages are experienced, health facilities borrow drugs from each other, or patients are referred to other health facilities.

> "*Mostly we are supplied through the county, by the facilitation of the NTD County Coordinator.*" (KII_HCW_R009_LSCH).

> "*Stock management is departmental...so we have those commodities that are for diagnosis …yes we have those commodities that are for treatment… so for us, we manage stock that is related to diagnosis.*" (KII_HCW_R001_LCRH).

> "*On stock management, sometimes when we have less stock, we borrow from Kakuma Mission Hospital*" (KII_HCW_R006_KSCH).

**Table 6. Monthly summary of leishmaniasis cases per facility surveyed in Turkana County between the year 2018 to 2022.**

| Summary cases | Overall | Monthly summary of cases per year | | | | |
|---|---|---|---|---|---|---|
| | | 2018 | 2019 | 2020 | 2021 | 2022 |
| **Overall** | | | | | | |
| Suspected cases | 1,640 | 345 | 249 | 224 | 266 | 424 |
| Positive cases | 632 | 86 | 77 | 88 | 117 | 202 |
| Relapse cases | 7 | 4 | 0 | 0 | 0 | 3 |
| Death cases | 18 | 0 | 0 | 18 | 0 | 0 |
| **Facility** | | | | | | |
| Kakuma SCH | | | | | | |
| Suspected cases | 227 | MR | MR | 11 | 36 | 104 |
| Positive cases | 100 | MR | MR | 5 | 13 | 50 |
| Relapse cases | 0 | MR | MR | 0 | 0 | 0 |
| Death cases | 9 | MR | MR | 9 | 0 | 0 |
| Lodwar CRH | | | | | | |
| Suspected cases | MR | MR | MR | MR | MR | MR |
| Positive cases | MR | MR | MR | MR | MR | MR |
| Relapse cases | MR | MR | MR | MR | MR | MR |
| Death cases | MR | MR | MR | MR | MR | MR |
| Lopiding SCH | | | | | | |
| Suspected cases | 676 | 105 | 86 | 112 | 151 | 182 |
| Positive cases | 229 | 24 | 30 | 28 | 58 | 70 |
| Relapse cases | 3 | 0 | 0 | 0 | 0 | 3 |
| Death cases | 9 | 0 | 0 | 9 | 0 | 0 |
| Namoruputh PAG HC | | | | | | |
| Suspected cases | 737 | 240 | 163 | 101 | 79 | 138 |
| Positive cases | 303 | 62 | 47 | 55 | 46 | 82 |
| Relapse cases | 4 | 4 | 0 | 0 | 0 | 0 |
| Death cases | 0 | 0 | 0 | 0 | 0 | 0 |

MR: Indicates that there were missing records for the specific years in the particular facility

**Table 7. Monthly summary of leishmaniasis commodity stock management per facility surveyed in Turkana County between the year 2018 to 2022.**

| Commodity summary | Overall | | |
|---|---|---|---|
| | Beginning Balance | Remaining Balance | Days of Stock Out |
| **Overall** | | | |
| Ambisome (50 mg vials) | 13 | 13 | 39 |
| Glucantime (5 ml vials)* | 0 | 0 | 30 |
| Paromomycin (2 ml ampule) | 2,440 | 1,675 | 68 |
| Sodium stilbogluconate (30 ml vials) | 2,253 | 1,690 | 61 |
| Miltefosine (50 mg caps)* | 0 | 0 | 30 |
| IT-Leish Bio RAD rK39 (piece) | 1,330 | 1,343 | 39 |
| DAT freeze-dried (5 ml vials) | 0 | 0 | 30 |

*These medicines are currently not in use in Kenya

Reports from the CHMT members indicated that VL commodities were exclusively donated by partners who are non--governmental organizations.

*"So the supplies are usually received through donations, we don't buy the supplies. We either get them "aah" because we have several donors giving us the medicines. We don't buy them from our county budget, so we get them as donations. There are usually reports we sign monthly and then we get the re-supplies. We can also get adjustments to the supplies when we are critically low and we haven't received our donations. We get that from our partners, there is a time one of the partners donated for us some supplies, and we also outsource from our neighboring county of West Pokot"* (KII_CHMT_R001)

### 3.6  Health facility preparedness in handling visceral leishmaniasis

From the qualitative data, majority of the health workers noted that they were well-capacitated to handle VL services within their facilities. The health workers felt that the decentralization model has led to accurate diagnosis of VL. The model has also led to improvement of the infrastructure within the health facilities, e.g., Lopiding Sub-County Hospital, which is now a referral facility for VL management. However, they decried stockouts, lack of enough personnel, inadequate training on VL management, and inability to offer blood transfusion services. Currently, only Lodwar County Referral Hospital offers blood transfusion services, which is an integral service for VL management.

*"We are okay, we are well equipped like right now. So in terms of any case that may come, be it a severe VL case we are fine, only those cases that may require blood transfusion we are not able to offer such services from our end because we don't transfuse blood here. We are supposed to send them to Lodwar County Referral Hospital or Lorugum, our, Sub-County Hospital. It becomes more strenuous for the patients when they are referred far away for blood transfusion"* (KII_HCW_R003_NPHC).

*"Insufficient or inadequate number of health care workers, becomes a challenge because we need more human resources to handle kala-azar patients together with other patients that we normally receive at the facility"* (KII_HCW_R016_LCRH).

## 4.  Discussion

We implemented the study to assess the effects of a decentralised model of VL treatment and care in Turkana County, Kenya. The decentralised model sought to expand services for VL patients among the marginalised and burdened communities in Turkana County. The services offered by this model included VL diagnosis, treatment, and follow-up to ensure successful treatment outcomes. Before the decentralization model was implemented in 2018, only six facilities in the county had the capacity to diagnose and treat VL patients. This situation was improved by the implementation of this model which saw the number of facilities increased to twenty-two by 2022. The assessment was based on a robust review of VL patients' health records from 2018 to 2022, and qualitative interviews with patients with VL, healthcare workers treating these patients, and CHMT members implementing this model. In our view, this robust assessment will help the VL elimination program in the country and the region to holistically address the policy gaps in VL treatment and care.

The study found that majority of the patients took over a month before visiting the facility to seek treatment and care after developing symptoms of VL. Patients attending Kakuma Sub-County Hospital reportedly took longer to seek treatment compared to those that attended the other three facilities. From the qualitative data, patients confirmed that the distance to the health facility was the biggest hindrance to them seeking care, lack of adequate knowledge on VL associated symptoms, poverty, and socio-cultural practices that subject VL patients to traditional treatment practices that

often lead to serious complications. Other studies done in VL endemic areas in Kenya have shown similar factors to be associated with late health-seeking behaviours [15]. According to healthcare workers and the CHMT members interviewed, some of the key barriers that prevent patients from seeking immediate care included negative attitudes towards the health facilities due to the perceived high rate of mortality in these public health facilities, and frequent insecurity cases in the region.

The study revealed that it took on average 11 days for patients to be initiated on treatment after diagnosis. This delay was occasioned by the need for referral of patients to better facilities that would first manage the severe cases that required blood transfusion and treatment of other opportunistic infections before being initiated on VL treatment. Further, this delay in the initiation of treatment was caused by stockouts of the key commodities used in VL diagnosis and treatment. For some of these key commodities such as combination therapy of SSG and PM, and RDT kits, the observed stockout periods were up to over 2 months. The healthcare workers reported that drugs are supplied to the health facilities through the county NTD department, this process sometimes results in delays occasioned by the county procedures and bureaucracies. The conversation with CHMT members indicated that the county receives VL commodities exclusively from different donors and partners on a regular basis. However, they complained of frequent stockouts of key commodities forcing them to sometimes borrow from neighbouring counties. For effective management of VL and long term sustainability, the key VL commodities should be budgeted for by the county governments where VL is endemic. Furthermore, in order to develop a feasible VL elimination road map for the region, all stakeholders including political leadership, and the donors should be mobilised and aligned during all phases of the elimination plan [16].

Notably, nearly all the patients that were initiated on treatment completed their treatment schedule with a successful outcome that saw over 98% of the patients initially cured. This is a testament of the positive outcome of the decentralization model in improving patient treatment outcome. The patients reported good experiences with the decentralized model, they noted general improvement in their health and well-being after they were initiated on treatment. The improvements were in terms of gains in body strength and weight, improved appetite, and cessation of epistaxis. This resulted in the resumption of normalcy for the patients. The conversation with healthcare workers appraised the positive impact of this model in improving and enhancing VL diagnosis and treatment in these marginalised communities. Due to this successful treatment outcome observed, the healthcare workers noted that other patients from neighbouring countries such as South Sudan and Uganda were coming to seek treatment from their facilities. This successful treatment outcome, from the view of CHMT members, was successful because of the enhanced support, and supply of VL commodities by partners.

Finally, the study found low rates of follow-up for VL patients. Only 2.5% of the patients put on treatment came for the six months' follow-up, with all showing a final definitive cure. Apart from confirming the final cure, follow-up is important as more cases of PKDL can be detected when there is active follow-up of patients, as reported in a recent study conducted in the neighbouring county of Baringo [17]. The main barrier reported by patients that hindered them from attending the six months follow-up visit was the associated high cost of transport to the health facilities. The distance to the health facilities was also a key barrier to effective follow-up of VL patients in endemic areas [18]. Again, majority of the patients also felt comfortable that they had been told that they were initially cured of VL, and therefore did not see the need to turn up for final follow-up visits. The healthcare workers indicated that they do not do active follow-up of patients mainly because of the lack of a follow-up system. They merely wait for treated patients to voluntarily and passively turn-up for the final six month's follow-up visits. However, for Kenya to achieve the VL elimination targets, active follow-up will be necessary in order to detect and treat PKDL cases so as to interrupt the disease transmission cycle [19]. Owing to this low follow-up rate witnessed in this study, there is therefore need for a strong follow-up mechanism to be put in place by the VL program in Kenya. Incidentally, the study established that three of the four facilities sampled in this study were managing VL patients as outpatients. This outpatient treatment model for VL needs to be studied in all the health facilities in Turkana County that provide this form of treatment, in order to test its cost-effectiveness.

## 4.1 Study strengths and limitations

The key strength of this study is that it is among the few studies to evaluate the expanded decentralized model for VL care in Kenya. The findings reported here are critical to influencing the improvement of the policy gaps in VL management and elimination. However, the study was not designed to collect detailed costing data of the model to robustly evaluate its cost-effectiveness and sustainability. We only sampled 4 out of the 22 facilities, which were serving as both diagnostic and treatment centres, thus limiting a detailed cost-effectiveness assessment of the model. Missing data from the records were observed, that limited accurate estimation of some of the key indicators. Additionally, this study only used descriptive methods to assess the effect of the decentralization model, thus limiting the ability to statistically determine other factors that influenced the treatment outcomes.

## Conclusions

The assessment of this model revealed that the initial cure rate was over 98%, however, with a low follow-up rate of only 2.5%. Further, the study county heavily relied on donor support thus limiting the efforts for sustainable VL elimination. For effective management and elimination of VL in Kenya, increased resource allocation from both the national and county governments is crucial. The study observed low knowledge and awareness among patients, which contributed heavily to the delayed time to diagnosis. This calls for revamped health education and awareness campaigns among the communities living in endemic areas to promote positive behaviour change for effective control of the disease. Due to low follow-up rate observed, there is a need for a strong follow-up mechanism to ensure the detection of early relapse cases.

## Acknowledgments

We would like to extend our appreciation to the following individuals and organizations for their invaluable support and contributions to this project, the Director General KEMRI, for the support throughout the project, the Turkana County government, and its respective employees who supported this study. We thank the patients and all the participants who volunteered their time to contribute to this study. Finally, we sincerely thank Jaspher Ndege, the study administrator for his logistical support, and Omar Kopi, who supported the training of the quantitative research assistants.

## Author contributions

**Conceptualization:** Jane Mbui, Dawn Maranga.

**Data curation:** Jane Mbui, Mariam Macharia, Collins Okoyo.

**Formal analysis:** Jane Mbui, Mariam Macharia, Collins Okoyo.

**Funding acquisition:** Jane Mbui.

**Investigation:** Jane Mbui, Mariam Macharia.

**Methodology:** Jane Mbui, Mariam Macharia, Collins Okoyo.

**Project administration:** Jane Mbui.

**Resources:** Jane Mbui, Dawn Maranga.

**Software:** Collins Okoyo.

**Supervision:** Jane Mbui, Mariam Macharia, Collins Okoyo.

**Validation:** Jane Mbui, Collins Okoyo.

**Writing – original draft:** Jane Mbui, Mariam Macharia, Collins Okoyo.

**Writing – review & editing:** Jane Mbui, Mariam Macharia, Dawn Maranga, Collins Okoyo.

## Appendices

### Appendix 1: Coding frame work for qualitative data analysis

| VARIABLE | CODE ASSIGNED |
|---|---|
| LODWAR COUNTY REFERRAL HOSPITAL | LCRH |
| NAMORPUTH PAG HEALTH CENTER | NPHC |
| LOPIDING SUB-COUNTY HOSPITAL | LSCH |
| KAKUMA SUB-COUNTY HOSPITAL | KSCH |
| PATIENT IN-DEPTH INTERVIEWS | IDI _ Patient_ Assigned respondent number_Facility code |
| CAREGIVER INDEPTH INTERVIEWS | IDI_Caregiver_Assigned respondent number_Facility code |
| HEALTH CARE WORKERS INTERVIEWS | KII_HCW_Assigned respondent number_Facility code |
| COSTING SURVEY INTERVIEWS | KII_ CHMT_Assigned respondent number |

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
