## [Decision Letter · Decision Letter 0]

15 Oct 2024

PONE-D-24-30004Assessment of a decentralization model in improving treatment and care of visceral leishmaniasis in Turkana County, Kenya: A mixed method studyPLOS ONE

Dear Dr. Okoyo,

Thank you for submitting your manuscript to PLOS ONE. After careful consideration, we feel that it has merit but does not fully meet PLOS ONE’s publication criteria as it currently stands. Therefore, we invite you to submit a revised version of the manuscript that addresses the points raised during the review process.

We look forward to receiving your revised manuscript.

Kind regards,

Mehdi Bamorovat

Academic Editor

PLOS ONE

Journal Requirements:

2. We note that Figure 1 in your submission contain [map/satellite] images which may be copyrighted. All PLOS content is published under the Creative Commons Attribution License (CC BY 4.0), which means that the manuscript, images, and Supporting Information files will be freely available online, and any third party is permitted to access, download, copy, distribute, and use these materials in any way, even commercially, with proper attribution. For these reasons, we cannot publish previously copyrighted maps or satellite images created using proprietary data, such as Google software (Google Maps, Street View, and Earth). For more information, see our copyright guidelines: http://journals.plos.org/plosone/s/licenses-and-copyright .  

3. In this instance it seems there may be acceptable restrictions in place that prevent the public sharing of your minimal data. However, in line with our goal of ensuring long-term data availability to all interested researchers, PLOS’ Data Policy states that authors cannot be the sole named individuals responsible for ensuring data access (http://journals.plos.org/plosone/s/data-availability#loc-acceptable-data-sharing-methods ).

Reviewers' comments:

Reviewer's Responses to Questions

**Comments to the Author**

1. Is the manuscript technically sound, and do the data support the conclusions?

Reviewer #1: Yes

Reviewer #2: Yes

2. Has the statistical analysis been performed appropriately and rigorously? 

Reviewer #1: I Don't Know

Reviewer #2: No

3. Have the authors made all data underlying the findings in their manuscript fully available?

Reviewer #1: Yes

Reviewer #2: Yes

4. Is the manuscript presented in an intelligible fashion and written in standard English?

Reviewer #1: Yes

Reviewer #2: Yes

5. Review Comments to the Author

Reviewer #1: This manuscript "Assessment of a decentralization model in improving treatment and care of visceral

leishmaniasis in Turkana County, Kenya: A mixed method study" by Mbui et al. is very interesting. The authors evaluated the impact of the decentralization of VL services in the city of Turkana, Kenya to inform policy at the county and national levels using a mixed methods cross-sectional survey conducted in four health facilities. The authors demonstrated that teaching the communities about the symptoms is essential to reduce delays in looking for reatment. Unfortunately, the authors did not try to replicate the study in another county.

Reviewer #2: 1. Title need to be precise as per research findings.

2. Abstract should more focused as per the findings not in general information.

3. Methodology should be more precise and well described of the survey process.

4. Result should well describe in diagnosis, treatment and follow up of visceral leishmaniasis.

5. Clarity of Figure 1 is very less, it need to be change with a good picture.

6. Legends to the figures and tables should be self-explanatory.

7. The discussion need to be revised to include a comprehensive analysis with critical examination of hypotheses. A more nuanced discussion of the implications of findings for public health and policy would strengthen this section.

8. In the Discussion section author should mention about the importance of the study.

9. Reference list need to be updated

10. Typological errors need to be corrected throughout the manuscript.

11. Scientific names should be in proper form throughout the manuscript.

12. The study needs a general revision in terms of language and grammar.

6. PLOS authors have the option to publish the peer review history of their article (what does this mean? ). If published, this will include your full peer review and any attached files.

**Do you want your identity to be public for this peer review?** For information about this choice, including consent withdrawal, please see our Privacy Policy .

Reviewer #1: **Yes: ** Rajendranath Ramasawmy

Reviewer #2: No

---

## [Author Response · Author response to Decision Letter 0]

18 Dec 2024

PONE-D-24-30004

Assessment of a decentralization model in improving treatment and care of visceral leishmaniasis in Turkana County, Kenya: A mixed method study

PLOS ONE

Dear Dr. Okoyo,

Thank you for submitting your manuscript to PLOS ONE. After careful consideration, we feel that it has merit but does not fully meet PLOS ONE’s publication criteria as it currently stands. Therefore, we invite you to submit a revised version of the manuscript that addresses the points raised during the review process.

We look forward to receiving your revised manuscript.

Kind regards,

Mehdi Bamorovat

Academic Editor

PLOS ONE

Journal Requirements:

>> We have checked and ensured that our manuscript meets PLOS ONE’s style requirements.

2. We note that Figure 1 in your submission contain [map/satellite] images which may be copyrighted. All PLOS content is published under the Creative Commons Attribution License (CC BY 4.0), which means that the manuscript, images, and Supporting Information files will be freely available online, and any third party is permitted to access, download, copy, distribute, and use these materials in any way, even commercially, with proper attribution. For these reasons, we cannot publish previously copyrighted maps or satellite images created using proprietary data, such as Google software (Google Maps, Street View, and Earth). For more information, see our copyright guidelines: http://journals.plos.org/plosone/s/licenses-and-copyright.

>> The original map in Figure 1 has been deleted and another one provided which is now showing the selected study health facilities. The map was created using ArcGIS Desktop version 10.2.2 software (Environmental Systems Research Institute Inc., Redlands, CA, USA). The base layer of the map was obtained from Environmental Systems Research Inc., (ESRI) (https://www.esri.com/en-us/arcgis/products/arcgis-online/features/make-maps).

3. In this instance it seems there may be acceptable restrictions in place that prevent the public sharing of your minimal data. However, in line with our goal of ensuring long-term data availability to all interested researchers, PLOS’ Data Policy states that authors cannot be the sole named individuals responsible for ensuring data access (http://journals.plos.org/plosone/s/data-availability#loc-acceptable-data-sharing-methods).

>> We have moved the minimal anonymized data to the GitHub public repository link, see (https://github.com/mancollo/VL-Decentralisation-Paper).

Reviewers' comments:

Reviewer's Responses to Questions

Comments to the Author

1. Is the manuscript technically sound, and do the data support the conclusions?

Reviewer #1: Yes

Reviewer #2: Yes

2. Has the statistical analysis been performed appropriately and rigorously?

Reviewer #1: I Don't Know

Reviewer #2: No

3. Have the authors made all data underlying the findings in their manuscript fully available?

Reviewer #1: Yes

Reviewer #2: Yes

4. Is the manuscript presented in an intelligible fashion and written in standard English?

Reviewer #1: Yes

Reviewer #2: Yes

5. Review Comments to the Author

Reviewer #1:

This manuscript "Assessment of a decentralization model in improving treatment and care of visceral leishmaniasis in Turkana County, Kenya: A mixed method study" by Mbui et al. is very interesting. The authors evaluated the impact of the decentralization of VL services in the city of Turkana, Kenya to inform policy at the county and national levels using a mixed methods cross-sectional survey conducted in four health facilities. The authors demonstrated that teaching the communities about the symptoms is essential to reduce delays in looking for reatment. Unfortunately, the authors did not try to replicate the study in another county.

>> The decentralization model was first implemented in Turkana County for a period of 5 years (2018-2022) by an NGO called FIND. Only Turkana County has implemented this model for five years. Therefore, this current study sought to assess the impact of this model in this county in expanding VL services to the marginalized affected communities that are far from the main health facilities. The implementation of the model was conducted independently by FIND, while the authors are simply assessing the effects of the expansion. With these results, the decentralization model can be adopted in other endemic counties in Kenya.

Reviewer #2:

1. Title need to be precise as per research findings.

>> The study assessed several parameters related to the decentralization model. The results of this assessment have been clearly outlined in the results section. The title is focused on the main aim of the study, which was to assess the decentralization model.

2. Abstract should more focused as per the findings not in general information.

>> The abstract has clearly outlined the key findings of this paper under the results subsection. We have slightly, added more key results in this subsection, see lines 39-40.

3. Methodology should be more precise and well described of the survey process.

>> The design of the study followed a secondary data abstraction and analysis process for the quantitative data. The qualitative data was collected using clearly defined data collection guides, which included in-depth interviews (IDIs) and key informant interviews (KIIs). The survey process for both quantitative and qualitative data collection has been clearly described in the methods section.

4. Result should well describe in diagnosis, treatment and follow up of visceral leishmaniasis.

>> The results section clearly describes the number of patient files abstracted (lines 215-221), VL cases diagnosed (subsection 3.3), and patients treated and followed up (subsection 3.4). Further, in the introduction, we have mentioned how VL is diagnosed and treated in Kenya (lines 74-91).

5. Clarity of Figure 1 is very less, it need to be change with a good picture.

>> We have deleted the original Figure 1 and provided another figure, which is now clear. The new figure is, however, showing the locations of the study health facilities.

6. Legends to the figures and tables should be self-explanatory.

>> We have revised the legends of the figures and tables to make them explicit and self-explanatory.

7. The discussion need to be revised to include a comprehensive analysis with critical examination of hypotheses. A more nuanced discussion of the implications of findings for public health and policy would strengthen this section.

>> The study is a descriptive secondary analysis that had no hypotheses stated. The discussion has been well focussed to state the public health and policy implications of the decentralization model, see lines 436-438, 465-467, 493-496, and 508-509.

8. In the Discussion section author should mention about the importance of the study.

>> The importance has been clearly started in lines 436-438.

9. Reference list need to be updated

>> The current references are fairly recent, ranging from 2010 to 2024. Importantly, VL being a neglected disease, not many studies are evaluating VL programmes in East Africa and the larger Africa region.

10. Typological errors need to be corrected throughout the manuscript.

>> We have thoroughly reviewed the manuscript to identify any typographical mistakes. Additionally, we passed the paper to a third independent reviewer for copy-editing.

11. Scientific names should be in proper form throughout the manuscript.

>> We have corrected scientific names throughout the manuscript.

12. The study needs a general revision in terms of language and grammar.

>> We have thoroughly reviewed the manuscript to identify any typographical, language and grammatical mistakes. Additionally, we passed the paper to a third independent reviewer for copy-editing.

6. PLOS authors have the option to publish the peer review history of their article (what does this mean?). If published, this will include your full peer review and any attached files.

Do you want your identity to be public for this peer review? For information about this choice, including consent withdrawal, please see our Privacy Policy.

Reviewer #1: Yes: Rajendranath Ramasawmy

Reviewer #2: No

While revising your submission, please upload your figure files to the Preflight Analysis and Conversion Engine (PA

---

## [Decision Letter · Decision Letter 1]

19 Feb 2025

PONE-D-24-30004R1Assessment of a decentralization model in improving treatment and care of visceral leishmaniasis in Turkana County, Kenya: A mixed method studyPLOS ONE

Dear Dr. Okoyo,

Thank you for submitting your manuscript to PLOS ONE. After careful consideration, we feel that it has merit but does not fully meet PLOS ONE’s publication criteria as it currently stands. Therefore, we invite you to submit a revised version of the manuscript that addresses the points raised during the review process.

We look forward to receiving your revised manuscript.

Kind regards,

Mehdi Bamorovat

Academic Editor

PLOS ONE

Reviewers' comments:

Reviewer's Responses to Questions

**Comments to the Author**

1. If the authors have adequately addressed your comments raised in a previous round of review and you feel that this manuscript is now acceptable for publication, you may indicate that here to bypass the “Comments to the Author” section, enter your conflict of interest statement in the “Confidential to Editor” section, and submit your "Accept" recommendation.

Reviewer #1: All comments have been addressed

Reviewer #3: (No Response)

2. Is the manuscript technically sound, and do the data support the conclusions?

Reviewer #1: Yes

Reviewer #3: Partly

3. Has the statistical analysis been performed appropriately and rigorously? 

Reviewer #1: Yes

Reviewer #3: No

4. Have the authors made all data underlying the findings in their manuscript fully available?

Reviewer #1: Yes

Reviewer #3: Yes

5. Is the manuscript presented in an intelligible fashion and written in standard English?

Reviewer #1: Yes

Reviewer #3: Yes

6. Review Comments to the Author

Reviewer #1: I thank the authors for addressing all my comments. This work is interesting to see how patients perceived the disease and highlight the purpose of educating caregivers to be able to provide rapid care and treatment.

Reviewer #3: The authors’ responses to the editorial comments are noted. It is particularly noted that the data has been made accessible in the GitHub platform.

In the previous reviews, there were concerns with the rigor and appropriateness of the statistical analysis.

The response to a previous review question on the abstract is unsatisfactory. It might be better to have more details in the methods and results section rather than the background section of the abstract. The authors should consider reviewing the abstract further.

The response to Reviewer 2’s comment number 7 reveals a concern: In the absence of a hypothesis, it is important to have clearly stated research questions.

Methods

Study Design: Is there a specific reason for not involving sub-county health management teams in the KIIs?

Line 127 to 128: Is there a chance that some cases were counted twice (from stock management forms and from patient records alluded to earlier and outlined in lines 134 - 135 )?

Results

In Table 1, it is interesting that the county referral hospital had no cases from 2018 to 2020 then suddenly there were 91 cases in 2021. Any explanation for this steep rise?

Line 198: When reporting measures of central tendency, it is best to consider which one to report between SD and IQR, based on the value of the SD versus the mean. In this case, the IQR should have been reported because the data looks skewed.

Are there specific reasons for not conducting bivariate analysis and multivariable logistic regression analysis were not conducted? How do we assess the statistical significance of your findings?

A key limitation of your study is that it cannot tell us whether there were other factors at play (potential confounders). For example, in 2020 and 2021, COVID-19 played a major role in health-seeking behaviour. In addition, over time, most counties in Kenya have invested in more health facilities.

7. PLOS authors have the option to publish the peer review history of their article (what does this mean? ). If published, this will include your full peer review and any attached files.

**Do you want your identity to be public for this peer review?** For information about this choice, including consent withdrawal, please see our Privacy Policy .

Reviewer #1: **Yes: ** Rajendranath Ramasawmy

Reviewer #3: No

---

## [Author Response · Author response to Decision Letter 1]

24 Mar 2025

Review Comments to the Author

Reviewer #1: I thank the authors for addressing all my comments. This work is interesting to see how patients perceived the disease and highlight the purpose of educating caregivers to be able to provide rapid care and treatment.

Author response: Thank you for taking your time to review our manuscript and for your valuable comments that improved the presentation of this work.

Reviewer #3: The authors’ responses to the editorial comments are noted. It is particularly noted that the data has been made accessible in the GitHub platform.

In the previous reviews, there were concerns with the rigor and appropriateness of the statistical analysis.

The response to a previous review question on the abstract is unsatisfactory. It might be better to have more details in the methods and results section rather than the background section of the abstract. The authors should consider reviewing the abstract further.

Author response: Thank you for this comment, we have reviewed and revised the abstract appropriately. However, due to the word count restrictions in the abstract section, we were unable to include extensive details.

The response to Reviewer 2’s comment number 7 reveals a concern: In the absence of a hypothesis, it is important to have clearly stated research questions.

Author response: We have added the research questions appropriately. See lines 107-113.

Methods

Study Design: Is there a specific reason for not involving sub-county health management teams in the KIIs?

Author response: Thank you for the question. The study sought to interview key personnel at the county level who influence and implement policy decisions at the county level. This group is the county health management team.

Line 127 to 128: Is there a chance that some cases were counted twice (from stock management forms and from patient records alluded to earlier and outlined in lines 134 - 135 )?

Author response: The cases were not counted twice. Each of the four forms served different purposes. For instance, individual demographic and treatment details were captured in the case management form, laboratory diagnosis results for each individual was captured in the laboratory logs, monthly summaries of all individuals diagnosed and treated for VL in the facility were captured in the monthly summary logs, and all VL test kits and medications administered to the patients in a month in the facility were summarized in the monthly stock management log.

Results

In Table 1, it is interesting that the county referral hospital had no cases from 2018 to 2020 then suddenly there were 91 cases in 2021. Any explanation for this steep rise?

Author response: Thank you for pointing this one out. We have replaced zero with missing records (MR). To indicate that the facility did not have VL case management patient records for the specific years under review.

Line 198: When reporting measures of central tendency, it is best to consider which one to report between SD and IQR, based on the value of the SD versus the mean. In this case, the IQR should have been reported because the data looks skewed.

Author response: Despite the skewed nature of the data for some variables, we reported the mean throughout the descriptive analysis because it is more informative and we were not performing any statistical tests other than the descriptive statistics. Further, the range (minimum and maximum), is more informative than the IQR in this context.

Are there specific reasons for not conducting bivariate analysis and multivariable logistic regression analysis were not conducted? How do we assess the statistical significance of your findings?

Author response: As we had stated earlier, the study used descriptive analysis rather than inferential analysis to assess the variables of choice. The study did not have any hypothesis to be tested using the logistic regression models or any other inferential techniques.

A key limitation of your study is that it cannot tell us whether there were other factors at play (potential confounders). For example, in 2020 and 2021, COVID-19 played a major role in health-seeking behavior. In addition, over time, most counties in Kenya have invested in more health facilities.

Author response: Thank you for your comment. However, conforming to the scope of this study, we were not determining factors associated with health seeking behavior or their potential confounders. We have added this as a limitation. See lines 495-497.

---

## [Decision Letter · Decision Letter 2]

21 Apr 2025

Assessment of a decentralization model in improving treatment and care of visceral leishmaniasis in Turkana County, Kenya: A mixed method study

PONE-D-24-30004R2

Dear Dr. Okoyo,

We’re pleased to inform you that your manuscript has been judged scientifically suitable for publication and will be formally accepted for publication once it meets all outstanding technical requirements.

Kind regards,

Mehdi Bamorovat

Academic Editor

PLOS ONE

Additional Editor Comments (optional):

Reviewers' comments:

Reviewer's Responses to Questions

**Comments to the Author**

1. If the authors have adequately addressed your comments raised in a previous round of review and you feel that this manuscript is now acceptable for publication, you may indicate that here to bypass the “Comments to the Author” section, enter your conflict of interest statement in the “Confidential to Editor” section, and submit your "Accept" recommendation.

Reviewer #1: All comments have been addressed

Reviewer #3: All comments have been addressed

2. Is the manuscript technically sound, and do the data support the conclusions?

Reviewer #1: Yes

Reviewer #3: Yes

3. Has the statistical analysis been performed appropriately and rigorously? 

Reviewer #1: I Don't Know

Reviewer #3: Yes

4. Have the authors made all data underlying the findings in their manuscript fully available?

Reviewer #1: Yes

Reviewer #3: Yes

5. Is the manuscript presented in an intelligible fashion and written in standard English?

Reviewer #1: Yes

Reviewer #3: Yes

6. Review Comments to the Author

Reviewer #1: I have no comments to the authors. This work is interesting to see how patients perceived the disease.

Reviewer #3: Previous comments have been adequately addressed by the authors. It is noted that there is a need for more advanced studies on the drivers of visceral leishmaniasis outbreaks in Kenya, considering that there is an ongoing outbreak in the country at the time of this review.

7. PLOS authors have the option to publish the peer review history of their article (what does this mean? ). If published, this will include your full peer review and any attached files.

**Do you want your identity to be public for this peer review?** For information about this choice, including consent withdrawal, please see our Privacy Policy .

Reviewer #1: **Yes: ** Rajendranath Ramasawmy

Reviewer #3: No

---

## [Editor Report · Acceptance letter]

PONE-D-24-30004R2

PLOS ONE

Dear Dr. Okoyo,

I'm pleased to inform you that your manuscript has been deemed suitable for publication in PLOS ONE. Congratulations! Your manuscript is now being handed over to our production team.

Kind regards,

on behalf of

Dr. Mehdi Bamorovat

Academic Editor

PLOS ONE